# Cognitive Interventions for the Treatment of Insomnia or Poor-Quality Sleep in Community-Dwelling Older People: A Systematic Review and Meta-Analysis

**DOI:** 10.3390/healthcare13091078

**Published:** 2025-05-06

**Authors:** Laura Pilar de Paz-Montón, José Alberto Laredo-Aguilera, Juan Manuel Carmona-Torres

**Affiliations:** 1Facultad de Fisioterapia y Enfermeria, Universidad de Castilla-La Mancha, Avda. Carlos III s/n, 45071 Toledo, Spain; josealberto.laredo@uclm.es (J.A.L.-A.); juanmanuel.carmona@uclm.es (J.M.C.-T.); 2Grupo de Investigación Multidisciplinar en Cuidados (IMCU), Universidad de Castilla-La Mancha, Avda. Carlos III s/n, 45071 Toledo, Spain; 3Instituto de Investigación Sanitaria de Castilla-La Mancha (IDISCAM), 45004 Toledo, Spain

**Keywords:** elderly, sleep quality, insomnia, cognitive interventions

## Abstract

**Background:** Aging and its pathologies, particularly sleep problems, are increasingly affecting industrialized societies. This study aims to determine the effectiveness of different cognitive interventions for the treatment of insomnia or poor sleep quality in community-dwelling older people. **Methodology**: A systematic review was carried out from November 2023–July 2024 according to the standards of the Preferred Reporting Items for Systematic Review and Meta-Analyses in the databases. The following databases were consulted: Pubmed, Web of Science, and ClinicalTrials.gov. The studies included patients with sleep problems or insomnia over 60 years of age. To evaluate the quality of the studies, the Critical Appraisal Skills Program Spanish (CASPe) guide was used. **Results:** Nine clinical trials with intervention groups and control groups belonging to the last 10 years were selected. They were analyzed, and the results were verified via questionnaires, scales, sleep diaries, and objective measures. In general, the implementation of the interventions improved the quality of sleep and symptoms of insomnia. **Conclusions:** Cognitive interventions have been found to be safe and useful for the treatment of insomnia and poor sleep quality in older people. Furthermore, they are feasible in terms of cost effectiveness and can be easily implemented by primary care teams.

## 1. Introduction

Aging is defined as a series of continuous, progressive, irreversible, universal, heterogeneous, and biopsychosocial changes associated with the interaction of genetic or biological factors (molecular and cellular damage, oxidative processes, hormonal changes, neurological degeneration, etc.), social, and cultural factors (limited participation, retirement, changes in residence, death of friends, etc.) that occur in humans over time, particularly changes that become more visible after the age of 60 [1,2,3].

In this context, there is a progressive population aging trend, predominantly in industrialized countries, with significant changes also occurring in middle- and low-income countries [1,4], with projections indicating that the number of people over 80 years old will triple worldwide by 2050 [1]. For example, in the European Union, there is a high proportion of elderly people, and when this proportion is combined with low fertility rates, aging becomes a concern [4,5]. In 2020, 21% of the population was over 65 years of age, and this figure is expected to increase in the coming years [5]. With the increasing aging of the population, health systems will come under greater pressure and will therefore have to address potential health problems. Some of the main pathologies or dysfunctions associated with aging include the loss of autonomy and functionality related to frailty, falls, immune, sensory, or organ dysfunction, decreased appetite, delirious states, psychiatric disorders, energy loss, urinary incontinence, the emergence of geriatric syndromes [1,6], and, specifically, changes in sleep patterns [2], with significant alterations [7] or deterioration in quality and quantity [8].

In particular, sleep problems are among the main issues affecting elderly individuals, as they are associated with shorter rest periods and an increase in nighttime awakenings with age [2,6,7,8]. Additionally, rest is linked to good physical and mental health; therefore, its absence and disturbance cause emotional, physical, and cognitive problems [9]. It is also associated with increased mortality and morbidity [10]. Furthermore, the DSM-5 notes that insomnia is the most significant sleep problem, with incidence rates ranging from 30 to 48% among older adults [7]. This issue arises because, over time, sleep patterns and circadian rhythms slow down due to the presence of steroid hormones, body temperature, and melatonin, which significantly impacts women [8,11].

As highlighted by the World Health Organization, healthy aging is an important way to improve the quality of life of older people. This concept emerged in the 1990s as a process in which participation, health, and security opportunities are optimized to increase the quality of life of individuals as they age [12,13]. It is a perspective that should be promoted by all institutions, social and health organizations, and members of society regardless of age, health, or socioeconomic level [14], considering all areas of well-being: physical, emotional, environmental, social, occupational, economic, or spiritual [12,14,15]. Healthy aging is important because it helps individuals live a healthier, longer, and more independent life. It can also reduce the risk of chronic disease and help individuals recover more easily from an illness, reducing health and personal costs [16]. Healthy aging can be promoted by physical exercise, good nutrition, proper sleep hygiene, self-esteem, and recreation [3]. The WHO declared the decade from 2021 to 2030 as the decade of healthy aging, aiming at improving the quality of life of older people, their families, and their communities [17].

The main solution for treating insomnia in the population over 65 years of age is the consumption of psychotropic drugs in 43.1% of cases, with the most commonly used drugs being tranquilizers and antidepressants [18]. Furthermore, it is well known that the use of medication generates health expenditures, as well as dependence (especially benzodiazepines), and has excessive adverse effects on the population, especially the elderly population, due to their slower metabolism [19]. Currently, various therapies are being studied to find safe, effective, nonpharmacological interventions that improve sleep quality in older adults. In recent studies, cognitive or cognitive behavioral therapies, including musical or cognitive behavioral therapies, have shown sleep benefits [20,21]. Cognitive or cognitive behavioral therapies (CBTs) are psychotherapeutic approaches that focus on identifying and modifying negative thought patterns and problem behaviors by combining techniques to change distorted thoughts with strategies to modify unwanted behaviors, helping people cope better with problems related to sleep. Cognitive behavioral therapies based on mindfulness reduce ruminations and emotional reactivity and promote the reappraisal of important experiences, all of which produce a cognitive restructuring that facilitates sleep and, hence, its effectiveness as a nonpharmacological therapy [22]. The reduction in costs of these therapies is because they are carried out in groups, thus reducing the number of medical visits and medication consumption, since their effect lasts for a long period of time [20,21].

In recent years, various systematic reviews have been conducted on the relationship between active aging and physical activity [23,24,25]. However, to our knowledge, no updated systematic reviews have been conducted that analyze nonpharmacological interventions (such as cognitive interventions) for insomnia treatment and sleep quality improvement.

Therefore, the objective of this study is to determine the effectiveness of various cognitive interventions in improving sleep quality in older people who live in the community and suffer from insomnia or poor sleep quality.

## 2. Methodology

### 2.1. Sources of Information

A systematic review was carried out following the standards of the PRISMA statement (Preferred Reporting Items for Systematic Reviews and Meta-Analyses) [26]. This revision has been registered in PROSPERO with the registration number CRD42024496623 [27].

### 2.2. Research Question and Search Strategy

First, the research questions are presented in Table 1.

The age cut-off point was set at people over the age of 60 years, following the cutoff used by the WHO and the United Nations when discussing older people. Therefore, to have a broader view, people over 60 years of age are included [1,28].

The searches were carried out by consulting the PubMed, Web of Science, and Clinicaltrials.gov databases from November 2023 to April 2024. The search strategies were reviewed and updated in April 2025. The search strategies were carried out in the previously mentioned databases and are shown in Table 2.

The search was carried out using the Boolean operator “AND” combined with the MeSH terms included in DeCS [29].

### 2.3. Inclusion and Exclusion Criteria

The inclusion and exclusion criteria selected for the selection of articles are detailed in Table 3.

The exclusion of nursing homes highlights the need to evaluate the effectiveness of these interventions in older people who maintain a degree of independence in the basic and instrumental activities of daily living. In institutionalized settings, external factors such as constant medical care, structured routines, and greater functional dependence may influence intervention outcomes, making it difficult to extrapolate findings to the noninstitutionalized elderly population. In addition, older adults with a greater degree of autonomy tend to benefit more from strategies that require active involvement, such as cognitive behavioral therapy (CBT). However, for individuals with severe cognitive or physical impairment, these interventions may not be applicable or may require considerable adaptation, which affects the external validity of the results; for this reason, people with serious pathologies have not been included either [30].

In this systematic review, only experimental randomized trials (clinical trials) in which different interventions are analyzed to improve the quality of sleep are included, since they provide the most scientific evidence. The purpose of this methodology is to provide the maximum evidence and quality available, considering that the information is limited, as it is a topic that has not been widely studied in this age range.

### 2.4. Selection Process

The process of study selection was conducted collaboratively by LPPM and JALA, adhering to the established inclusion and exclusion criteria. Citations were organized, and duplicates were removed via the Mendeley reference manager. After duplicates were removed, titles and abstracts were reviewed to identify studies for in-depth examination. Following a full-text analysis, the final selection of studies for inclusion in the qualitative and quantitative synthesis was made. In cases where discrepancies arose during the selection process, discussions between LPPM and JALA were held, and if no consensus was reached, a third author (JMCT) was consulted to resolve disagreements and achieve a consensus.

### 2.5. Quality Assessment

To evaluate the different studies included in this systematic review, the critical appraisal skills program Spanish (CASPe) guide was used [31].

For the analysis of the methodological quality of the experimental studies included in this review, “11 questions to understand a clinical trial” from the CASPe Guide [31] was used. The first 6 questions were used to assess the validity of the study. The first 3 were elimination questions and examined whether the research question was well defined, and they analyze the follow-up and quality of the randomization of the sample. Questions 4, 5, and 6 are detailed and assess the similarity and follow-up of both the treatments and the groups. Questions 7 and 8 analyze the precision and effect of the treatment in terms of the results. Finally, questions 9, 10, and 11 are related to the benefits obtained from the clinical trial: applicability, cost–benefit, and consideration of all results with clinical relevance.

The questions must be answered with “yes”, “no”, or “I do not know”, considering this guide.

### 2.6. Data Extraction

Data extraction was performed via LPPM and JALA. The results were extracted via the Mendeley bibliographic manager and were manually classified and read the data comprehensively. The following has been taken into account:-Identification of the article using the title and summary.-Methodology: Type of intervention chosen and experimentality.-Participants: average age, control and intervention groups, number of participants, and inclusion criteria and selection.-Cognitive interventions, techniques, or therapies are applied to improve sleep quality.-The following variables were measured: quality and quantity of sleep, number of nocturnal awakenings, and time.-Results of the intervention: maintenance or improvement in the aspects measured with the variables.

### 2.7. Analysis of the Data Obtained

A qualitative synthesis was performed for each study in this review. They compared the 9 different therapies to the comparator group, placebo, or other applied techniques simultaneously in the second randomized group.

For quantitative synthesis, RevMan 5.4 software was used to perform different random effects meta-analyses to analyze the impact of cognitive interventions on sleep quality, the insomnia score index (ISI) score, and the Pittsburgh Sleep Quality Index (PSQI) score. Statistical heterogeneity was assessed via the I^2^ statistic and was defined as follows: I^2^ intervals ≤ 25%, 26–50%, or ≥51% indicated low, medium, or high heterogeneity, respectively. Statistical significance was set at 0.05, and publication bias was assessed using a visual inspection of the funnel plot.

## 3. Results

### 3.1. Study Selection and Characteristics

Three databases were consulted, from which, when the search strategy defined above was applied, a total of 1265 articles were obtained. Among these, 39 were duplicates, so 1126 articles were obtained. After the titles and abstracts were read, only 47 clinical trials were selected for analysis, and the full texts were read. Finally, a total of nine clinical trials [32,33,34,35,36,37,38,39,40] were included, with a total of 1079 participants, of which 641 were part of the intervention group and 438 were from the control or comparator groups. Figure 1 shows a PRISMA flow chart for the selection of systematic review studies.

All studies included older or elderly people with a mean minimum age of 60 years who had poor sleep quality or insomnia. Each intervention followed the established ethical standards, and all the interventions had favorable reports from the established ethics committees. Table 4 presents an analysis of the data and main characteristics of the studies included in the present systematic review.

### 3.2. Risk of Bias

The nine included studies were assessed for risk of bias via the CASPe guide for RCTs. Table 5 shows the evaluation of the different domains for each study.

The studies were considered of sufficient quality and valid when at least eight “yes” answers were given.

After the methodological quality of all the included clinical trials was evaluated, all of them were of considerable quality, and none of the studies were excluded because of low quality.

### 3.3. Sleep Measurements

#### 3.3.1. Sleep Quality and Cycle

Among the studies analyzed in the present systematic review, actigraphy or polysomnography was used in six of them to measure sleep quality and cycles [32,33,34,35,37,38,39,40]. It was generally observed that, after the application of the different interventions in the studies, there was an improvement in the time of wakefulness, latency period, and quality and quantity of sleep in the intervention group (IG) compared to the control group (CG).

Lovato et al. [32] reported that, after the application of cognitive behavioral therapy for insomnia (CBT-I), there was a significant reduction in wakefulness after the onset of sleep in the IG compared to the CG. Compared to that in the CG, the total sleep time in the IG significantly increased after the intervention. In Lovato et al. [34], a 4-week cognitive behavioral therapy program for insomnia was used. Polysomnography was used to classify people into short sleep (less than 6 h of sleep) and long sleep (more than 6 h of sleep) groups. Actigraphy was used for the measurement of sleep parameters and revealed that the duration of total sleep time was significantly shorter in the IG than in the CG after the CBT-I intervention was applied and at 3 months of follow-up. In the study by Bergdahl et al. [35], after applying the intervention consisting of CBT-I (cognitive behavioral therapy for insomnia) accompanied by a group manual, the IG woke up and rose earlier than before the intervention and spent less time in bed, but they managed to improve their sleep efficiency and decrease the sleep latency period that was maintained during the follow-up. In Dzierzewski et al. [38], a CBT-I intervention in the IG and a sleep education program in the CG were applied. A significant difference was observed in the IG, which had a shorter awakening duration and slept less time than the CG did. In Perini et al. [39], the IG significantly decreased awakening after sleep onset, and the rest of the group variables did not significantly differ. An intervention based on mindfulness measured with actigraphy was applied in the IG. In contrast, in the CG, an intervention based on educational talks about sleep guided by a clinical psychologist was applied.

A mixed-effects meta-analysis (Figure 2) was performed with six studies [32,33,34,35,37,39] on sleep quality, as measured using actigraphy and polysomnography. The meta-analysis revealed a standardized mean decrease in favor of the intervention group of −1.14 (95% CI: −3.88–1.6), with an I^2^ heterogeneity of 98%. The risk of publication bias for this meta-analysis is reflected in Figure 3.

#### 3.3.2. Sleep Diary

The use of a sleep diary was used in four of the studies analyzed [32,33,34,37,38]. It has been observed that, as with actigraphy and polysomnography, there is also a subjective improvement in insomnia symptoms in the studies analyzed.

Lovato et al. [32] reported a reduction in night awakenings and greater sleep efficiency in the IG than in the CG both after the intervention (CBT-I) and at follow-up. For Alessi et al. [33], a significant decrease in sleep onset latency was observed in the IG compared with the CG after the application of cognitive behavioral therapy. In the case of Lovato et al. [34], a significant subjective reduction in time awake after sleep was observed in the IG, and in the study by McCrae et al. [37], an individual behavioral sleep treatment program was applied to the GI tract, and the CG had weekly conversations with a therapist about non-sleep issues. A significant improvement in the IG was observed in sleep onset latency, awakening after sleep onset, and sleep efficiency, which was maintained at follow-up.

A mixed-effects meta-analysis (Figure 4) was performed with four studies [32,34,37,38] on subjective sleep quality, as measured using a sleep diary. The meta-analysis revealed a standardized mean decrease in favor of the IG of 5.51 (95% CI: 2.74, 8.28), with a I^2^ heterogeneity of 90%. The risk of publication bias for this meta-analysis is reflected in Figure 5.

#### 3.3.3. Scoring on the Insomnia Severity Index (ISI) Scale

The ISI scale was applied in five of the studies analyzed [32,33,34,39,40]. It can be concluded that, after the application of the different interventions, there was a reduction in insomnia rates in the IG compared with those in the CG.

Lovato et al. [32] reported that insomnia rates were significantly lower (by 6.94 points) in the IG than in the CG and were maintained during follow-up, with a difference of 6.01 points. In the study by Alessi et al. [33], there was a reduction in the score of at least one point in the IG. Lovato et al. [34] reported a significant reduction in insomnia indices in the IG, which was maintained throughout the follow-up period. In the short sleep group, there was a difference of 4.11 points after the intervention compared with the CG and 4.73 points at the 3-month follow-up. In the long sleep group, the difference was 4.82 points after the intervention was applied in the IG compared with the CG and 4.4 points at 3 months of follow-up. In the case of Perini et al. [39], there was a reduction in the score in both the CG and IG after the intervention, but a greater reduction in the severity of insomnia was evident in the IG, with a significant difference of 1.28 points in the IG compared to the CG after the intervention. In the study by Camino et al. [40], participants were divided within their respective IG and CG into subclinical (those with symptoms of insomnia but undiagnosed) and moderate (diagnosed insomnia) groups, and mindfulness-based therapy was applied in the IG and film sessions on active aging in the CG. In the IG, a significant decrease in the ISI score was evident after the intervention in both the IGsubclinical and IGmoderate groups compared to the CG, with a score of approximately 3 points.

A mixed-effects meta-analysis (Figure 6) was performed with three studies [33,34,39] on sleep quality by the ISI. The meta-analysis revealed a standardized mean decrease in favor of an IF of −3.94 (95% CI: −5.61, −2.28), with an I^2^ heterogeneity of 98%. The risk of publication bias for this meta-analysis is reflected in Figure 7. The remaining studies could not be included because of a lack of reported data.

#### 3.3.4. Scoring on the Pittsburgh Sleep Quality Index (PSQI) Scale

In terms of the scores on the PSQI scale, in practically all of the five studies that included it, there was an improvement in the scores in the IG compared with those in the CG [33,36,38,39,40]. The higher the score on the scale, the worse the quality of sleep.

In the study by Alessi et al. [33], cognitive behavioral therapy was applied. A higher quality of sleep was observed in the IG than in the CG both after the intervention, where the IG score was 2.2 points lower than the CG score, and at the follow-up, where the IG score was 1.2 points lower than the CG score. In the case of Gallegos et al. [36], in the IG, mindfulness-based cognitive therapy was applied to help control breathing and stress, and in the CG, a placebo was applied. The score in the IG decreased significantly by approximately 1 point after the intervention was applied. In Dzierzewski et al. [38], a significant score of 2.1 points less was obtained in the IG than in the CG after the intervention was applied. In the study by Perini et al. [39], both the CG and IG reported improvements in the PSQI scale score after the intervention that were maintained during the follow-up, with a difference of 0.24 points in the IG with respect to the CG after the intervention and at 6 months of follow-up. In Camino et al. [40], the IG experienced a significant improvement in sleep quality with a decrease in PSQI score after the intervention of 3.45 points in the IG with respect to the CG.

A mixed-effects meta-analysis (Figure 8) was performed with three studies [33,36,39] on sleep quality, as measured using the PSQI scale. The meta-analysis revealed a standardized mean decrease in favor of an IG of −1.05 (95% CI: −2.49, 0.39), with a I^2^ heterogeneity of 99%. The risk of publication bias for this meta-analysis is reflected in Figure 9. The remaining studies could not be included because of a lack of reported data.

## 4. Discussion

This review highlights that cognitive interventions play a beneficial role in improving sleep quality in older people. This is important because, compared with pharmacological treatments, nonpharmacological treatments are not utilized often because of a lack of knowledge and awareness about their benefits.

To test the effectiveness of the interventions, objective and subjective instruments have been used. The ISI and PSQI subjective scales indicate that CBT-I and cognitive therapies based on mindfulness obtained similar results [32,33,34,35,36,37,38,39,40]. The purpose of cognitive therapies based on mindfulness is to replace maladaptive sleep habits and thoughts to reduce the arousal associated with sleep [22]. For objective measures, both actigraphy and polysomnography were used to record the number of awakenings, hours of sleep, and quality of sleep. Compared to objective methods, which are based on monitoring movements that occur during sleep or activity for prolonged periods of time, subjective methods, highlighted in a sleep diary, are usually slightly biased by the subjectivity of the measurements and the difficulty of correctly counting behavior [41]. These discrepancies in scoring can be observed in some of the studies included in the present systematic review [32,33,37], which may be due to the different stages of sleep, specifically the increase in slow waves or the REM phase, which are subjectively perceived as greater improvements in sleep than captured by actigraphy [42].

Returning to the previous point, pharmacological therapies play a fundamental role in the management of sleep quality; despite the progress of nonpharmacological therapies, they are not usually the first option for patients, owing to a lack of knowledge, as interventions are associated with less risk than the consumption of drugs (they do not interact with other medications), and they also help in the diagnosis of coexisting mental pathologies. Cognitive therapies provide more comprehensive care that suggests that they should be more accessible and implemented in the community with the possibility of group interventions. Therefore, there is a tendency to use psychotropic drugs because of the speed of effectiveness they produce; following cognitive therapy requires greater effort and time to achieve good adherence, although, at the level of economic cost, they are less expensive to apply, as they can be applied in a group, and once adherence is carried out, they achieve long-lasting results over time [43,44]. There are several clinical practice guidelines and protocols in Spain that describe the mode of action [45,46]. In addition, cognitive behavioral therapies are effective when there are other underlying pathologies leading to insomnia or poor sleep quality, such as mental or neurological pathologies [47,48].

Importantly, nonpharmacological therapies are currently being developed because of the benefits observed, so it is interesting to know which ones can improve sleep quality and which could be considered for future research. Some of them that show beneficial effects for improving sleep quality compared with pharmacological therapies are physical exercise [49], music therapy [50], or virtual reality [51,52]. The benefits of combining several nonpharmacological therapies are also discussed [53,54]. Physical exercise specifically improves insomnia if individuals carry out a training program with exercise guided by a qualified person [49]. Virtual reality interventions have also been shown to be effective and cost-effective, resulting in significant improvements in this problem [51,52], and a music therapy intervention in a population over 60 years has been proven that listening to relaxing music for at least 4 weeks before sleeping effectively improves sleep quality [50]. Therapies such as cognitive behavioral therapy with physical training or the combination of a guided exercise program with acupuncture therapy have also been shown to have a positive effect on sleep quality in both cases [53,54]. This field would be interesting to consider in future research in order to increase knowledge in this area.

Furthermore, it is important to train health personnel to be responsible for promoting nonpharmacological treatments, knowing the different changes in circadian rhythms in old age, how old age affects a decrease in nighttime sleep, and how older people experience an increase in daytime naps to be able to apply appropriate hygiene measures [55], rather than resorting to nonpharmacological interventions as a first option. Good health education is essential because, as previously described, there is a tendency to consume drugs instead of paying attention to environmental factors, such as the comfort of the bedroom, bed, temperature, nighttime noises, or sleep habits and routines (it is advisable to always go to bed and get up at the same time and only stay in bed for the necessary time to avoid sleep fragmentation), to address this problem, and, with the supervision of health professionals, better management, a reduction in insomnia rates and the consequent reduction in health expenditure in terms of pharmacological therapy would be possible [32,33,34,35,36,55,56].

### Limitations

Regarding the limitations of this study, it is worth highlighting the heterogeneity of the methods or scales used to assess sleep quality in the clinical trials included in this review. Another limitation would be the heterogeneity in the variables resulting from post-intervention measurements. On the other hand, the effectiveness of cognitive therapies for improving sleep in comparison with other techniques, such as physical activity, could not be verified, so it would be advisable to conduct more clinical trials in this sense. Another limitation is the heterogeneity of the therapies applied; this review did not separate the effects of different lengths of therapy or types of therapy. According to the results obtained in the meta-analysis, the 6-week CBT-I therapy obtained the best results, but further research is needed to reach a conclusion.

The age of the participants is another limitation, as the range for being considered an older person differs between countries. An analysis by age range is recommended for future research. In addition, only people living in a community were included, as this was the focus of this study, but it would be interesting for future research to include the institutionalized geriatric population and explore how these therapies can be adapted and help them improve their sleep.

A gender analysis would also be necessary in future studies to consider the diversity of cultures. Research indicates that insomnia is more common among older women than among men [57]. This is influenced by the different stages of a woman’s life that affect her sleep patterns. Women have longer sleep latency and greater daytime sleepiness. After CBT-I therapy, both sexes had similar results, but men suffered more from long-term consequences. Future research should delve deeper into these sex-specific factors to optimize treatment strategies for older adults with sleep disorders [58,59].

Notably, this is the first review to analyze the effects of cognitive interventions to improve sleep; all of the studies included in this review presented a low risk of bias, and all were clinical trials.

## 5. Conclusions

Specific cognitive interventions to improve sleep quality in elderly people in the community have proven to be effective, improving symptoms of sleep disturbances and insomnia. Given their simplicity, low resource requirements, and cost-effectiveness, particularly in groups, these cognitive behavioral interventions hold significant potential for integration into health services. Their application in primary care teams could increase accessibility, improve patient outcomes, and contribute to cost reduction in an aging population with increasing demands on the health system.

## Figures and Tables

**Figure 1 healthcare-13-01078-f001:**
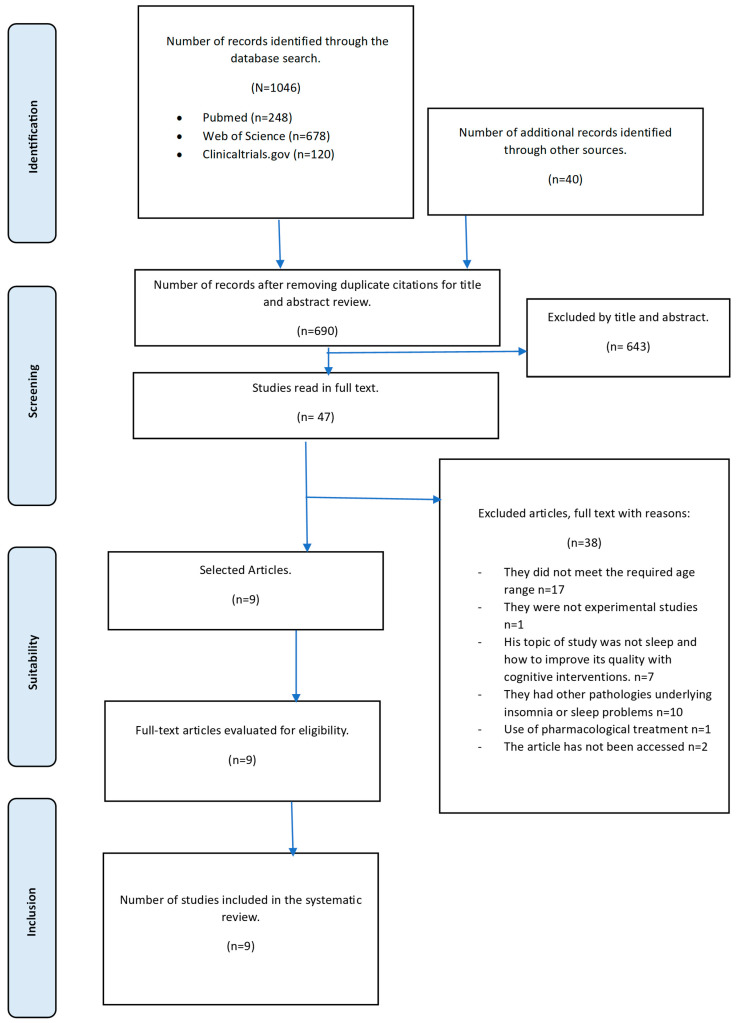
Flowchart for study selection.

**Figure 2 healthcare-13-01078-f002:**
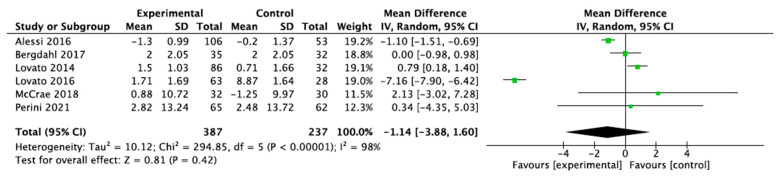
Sleep efficiency of actigraphy and polysomnography.

**Figure 3 healthcare-13-01078-f003:**
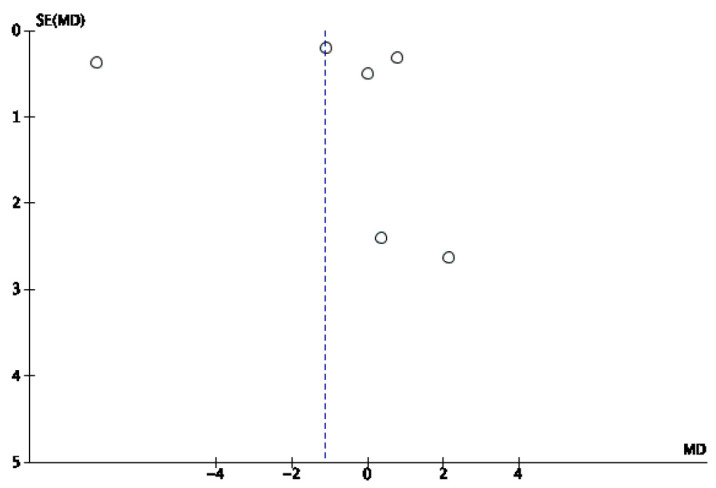
Funnel plot of sleep efficiency via actigraphy and polysomnography.

**Figure 4 healthcare-13-01078-f004:**
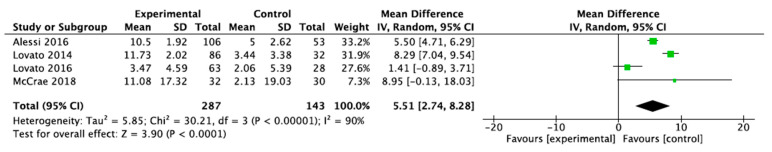
Sleep diary efficiency.

**Figure 5 healthcare-13-01078-f005:**
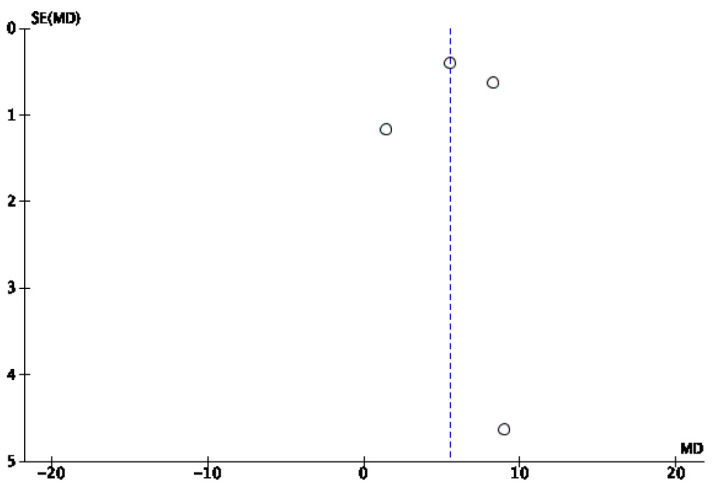
Forest plot of sleep diary efficiency.

**Figure 6 healthcare-13-01078-f006:**
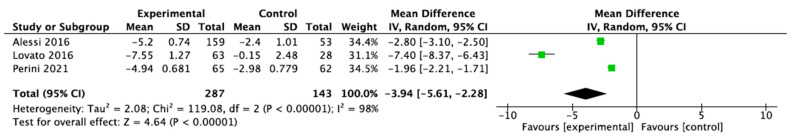
ISI efficiency.

**Figure 7 healthcare-13-01078-f007:**
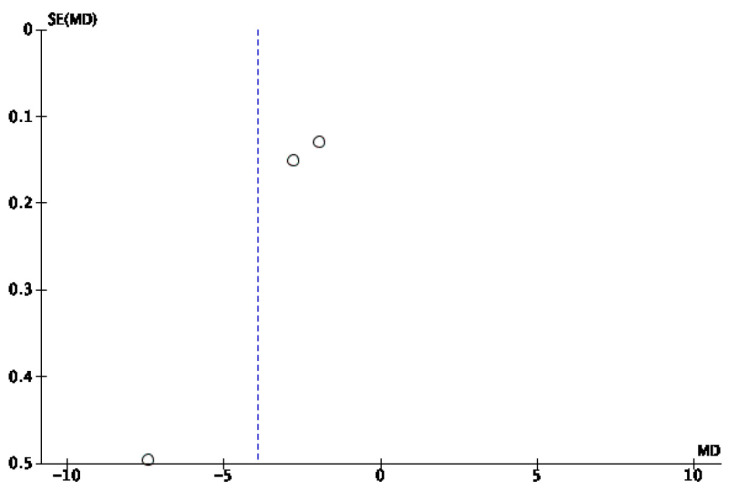
Funnel plot of ISI efficiency.

**Figure 8 healthcare-13-01078-f008:**
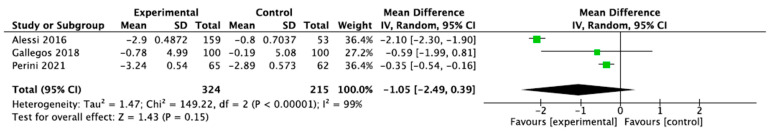
PSQI score efficiency.

**Figure 9 healthcare-13-01078-f009:**
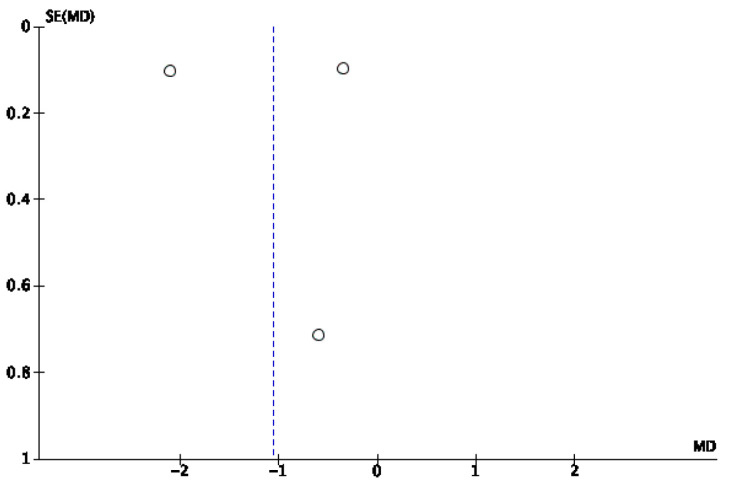
Funnel plot of PSQI score efficiency.

**Table 1 healthcare-13-01078-t001:** Research questions in the PICO format.

Do Cognitive Therapies, Interventions and Techniques Improve Sleep Quality in Older People?			
Population (P)	Intervention (I)	Comparison (C)	Outcome (O)
People over 60 years of age with poor sleep quality or insomnia.	Cognitive interventions, therapies, and techniques to improve sleep quality.	People over 60 years of age in whom interventions to improve sleep quality are not applied.	Increase and improvement in sleep quality in older people after the intervention.

**Table 2 healthcare-13-01078-t002:** Search strategy used for each database.

Database	Search Strategy
Pubmed	((cognitive[Title/Abstract]) AND (Aged[Title/Abstract]) OR (“older people”[Title/Abstract]) OR (elderly[Title/Abstract])) AND ((sleep[Title/Abstract]) OR (insomnia[Title/Abstract]))
Web of Science	(cognitive) AND (Aged OR elderly) AND (sleep OR insomnia)
Clinicaltrials.gov	“cognitive AND aged AND sleep OR insomnia”

**Table 3 healthcare-13-01078-t003:** Inclusion and exclusion criteria for articles on interventions, techniques, and therapies to improve sleep quality in older adults.

Inclusion Criteria	Exclusion Criteria
Type of studies: randomized experimental studies with a control group.	Studies with more than 10 years of experience.
Population over 60 years of age with worsening sleep quality or insomnia who do not suffer from other serious pathologies	Elderly population living in nursing homes.
Studies written in English.	Studies that analyze the effectiveness of physical or pharmacological interventions on sleep quality and that do not use cognitive interventions, therapies, or techniques to improve sleep quality.

**Table 4 healthcare-13-01078-t004:** Main characteristics of the studies included in this systematic review.

First Author, Year, Country	Design	Participants and Characteristics	Interventions and Follow-Up	Variables and Measures	Evaluation and Results	Losses to Follow-up, Adverse Effects and Limitations
Lovato et al. [32], 2014. Adelaide.	Randomized con-trolled clinical trial.	N = 118 (55 M, 66 W) IG: 86 CG: 32 A: 63.7 C: (1) AHI less than 15. (2) They do not suffer from mental disorders. (3) They have not consumed hypnotics or large amounts of caffeine for at least 1 month. (4) They have nocturnal awakenings of +30 min at least 3 nights per week for 6 months. (5) They suffer from fatigue or memory problems during the day.	IG: Brief CBT-I applied in 4 sessions of 60 min during 4 weeks in groups of 4 or 5 people. The first session dealt with the behavioral component, the 2nd and 3rd sessions dealt with sleep hygiene and education, and the 4th session summarized and reviewed what had been worked on. CG: The same follow-up as GI was carried out, but the intervention was not applied until after the study had finished.	Sleep diary: recorded hours of going to bed, lights out, getting out of bed, number and duration of awakenings, and estimated sleep latency. 7 days per assessment. Wrist actigraphy: sleep latency and number of awakenings. 7 days per assessment. ISI: changes in insomnia. Flinders Fatigue Scale, Epworth Sleepiness Scale, and Daytime Feeling and Functioning Scale: daytime performance.	Evaluation T1: during the intervention. T2: after the intervention. T3: 3 months post-intervention. Results Significant reduction in night awakenings and improvement in sleep efficiency after the application of the intervention in IG compared to CG.	IG: 2 did not receive interv, 14 lost in phone tracking. CG: 2 do not receive interv, 5 lost in phone tracking.
Alessi et al. [33], 2016. Los Angeles.	Randomized con-trolled clinical trial.	N = 159 IG: 106 (102 M, 4 W) CG: 53 (52 M, 1 W) A: 64–80 C: (1) +60 years. (2) Results higher than 24 in Mini Mental Test. (3) Do not present severe mental disorder. (4) AHI less than 20.	IG: Cognitive behavioral therapy was applied in 5 sessions of 1 h over 6 weeks + telephone call for control in week 5. Stimulus control, sleep restriction, cognitive therapy, sleep hygiene, and relapse prevention were treated. CG: General sleep education pro-gram at the same frequency and intervals as GI.	Sleep efficiency (sleep onset latency, total wake time, and sleep efficiency): diary recording. Sleep parameters: wrist actigraphy. Insomnia: ISI. Sleep quality: PSQI.	Evaluation T0: prior to application of the intervention. T1: one-week post-intervention. T2: 6 months post-randomization. T3: 12 months post-randomization. Results Statistically significant improvement in IG compared to CG in sleep quality and insomnia after application of the intervention. Effects lasting + 12 months.	Almost all the sample were men. IG: 9 refused the interview, 3 died, 2 impossible to contact, 3 withdrew from the study. CG: 1 withdrew, 1 impossible to contact.
Lovato et al. [34], 2016. Adelaide	Randomized con-trolled clinical trial.	N = 91 (43 M, 48 W) IG: 63 CG: 28 A: 63.34 C: (1) Waking up during the night for more than 30 min. at least 3 days a week for 6 months. (2) Fatigue, poor performance, and even memory problems during the day. (3) Not taking any drug treatment for at least 1 month prior to the intervention. (4) AHI less than 15. (5) Not having severe mental problems or consuming excessive caffeine.	IG: Divided into short sleepers (−6 h) and long sleepers (+6 h). A 60 min weekly session of CBT-I consisting of sleep restriction, cognitive and educational therapy aimed at addressing sleep misperception and cognitive-emotional aspects was administered for 4 weeks. CG: Received the intervention after completing the 3-month follow-up. The same divisions were made into short and long sleepers as in IG.	Polysomnography: classification into short or long sleep. Sleep diary and actigraphy: for 7 days per measurement to calculate quality, awakenings, duration of sleep. ISI: insomnia. Flinders Fatigue Scale, Epworth Sleepiness Scale and Daytime Feeling and Functioning Scale: performance during the day.	Evaluation T0: pre-interv. T1: post-interv. T2: 3 months post-interv. Results The intervention produced lasting improvements in subjective sleep quality, perceived insomnia severity, daily performance, and thoughts about sleep in IG compared to CG. The improvements were more significant in the short-sleeper group.	IG: 1 does not receive interv, 9 are not followed up. CG: 1 does not receive interv, 9 are not followed up
Bergdahl et al. [35], 2017. Uppsala.	Randomized con-trolled clinical trial.	N = 67 (9 M, 58 W) IG: 35 (5 M, 30 W) CG: 32 (4 M, 28 W) A = 60.5 C: (1) Having been using nonbenzodiazepine hypnotics at least 3 times a week for 6 months, but with persistent symptoms of insomnia, wanting to stop taking them and stopping them 5 days before the intervention. (2) Speak Swedish. (3) Have no mental pathologies.	IG: CBT-I + a group manual for the intervention. Weekly 90 min sessions for 6 weeks by clinical psychologists. CG: 2 weekly sessions for 4 weeks of auricular acupuncture.	Actigraphy: daily patterns, specifically sleep periods. Short Form-12: physical and mental health, quality of life.	Evaluation T0: pre-intervention T1: post-intervention T2: 6 months post-intervention. Results Insomnia symptoms were significantly reduced in IG after the intervention. In CG, the intervention was not effective in reducing insomnia.	Sample majority of women. IG: 6 declined the interview, 6 dropped the inteview. CG: 4 declined to participate.
Gallegos et al. [36], 2018. New York.	Randomized con-trolled clinical trial.	N = 200 (76 M, 124 W) GI: 100 (38 M, 62 W) GC: 100 (38 M, 62 W) A: 72.5 IG: 72 CG: 73 C: (1) E: +65 (2) Speak English (3) Not have serious mental pathologies. (4) Have a stable pharmacological treatment of antidepressants or anxiolytics for at least 8 weeks prior to the interval and stop taking them during the treatment.	IG: Weekly 120 min sessions for 8 weeks and one intensive 7 h session of mindfulness-based stress reduction therapy in groups of 15 to 20 participants. Sessions included breathing exercises, sitting meditation or body scanning followed by informal practice at home. CG: Placebo, no intervention was applied, only assessments were performed.	PSQI: sleep quality.	Evaluation T0: pre-interv. T1: 8 weeks post-interv. T2: 6 months post-interv. Results This study was not directly designed to improve sleep quality, but significant improvements in sleep quality were observed following implementation of the intervention.	Unknown.
McCrae, et al. [37], 2018. Gainesville.	Randomized con-trolled clinical trial	N = 62 (20 M, 42 W) GI: 32 (10 M, 22 W) GC: 30 (8 M, 22 W) A = 62.45 C: (1) +65 years. (2) Agreed to randomization and speak and write English. (3) Suffer from insomnia. (4) Have no serious mental illness. (5) Have not taken prescribed sleeping medication in the last month.	IG: 1 weekly 1 h session for 4 weeks of an individually delivered behavioral treatment pro-gram on sleep hygiene, stimulus control, sleep restriction, and relaxation. CG: 1 weekly 1 h session for 4 weeks where you discuss topics unrelated to sleep or the IG with a therapist.	Sleep diary and actigraphy: for 14 days per assessment to measure quality, duration of sleep, night awakenings, etc. Geriatric Depression Scale, Beck Depression Inventory, Second Edition, State-Trait Anxiety Inventory-Formula, and Neuropsychological Battery: Mood and Depression.	Evaluation: T0: pre-interv. T1: during interv. T2: 3 months post-interv. Results: The IG experienced significant improvements in sleep measures that continued 3 months post-intervention. The CG did not experience significant improvements.	Unusually, 60% of older adults had higher education. IG: 5 did not receive interv, 8 lost to follow-up. CG: 7 did not receive interv, 2 lost to follow-up.
Dzierzewski et al. [38], 2019. Los Angeles.	Randomized con-trolled clinical trial.	N = 159 (154 M, 5 W) IG: 106 (102 M, 4 W) -52: CBTI group -54: CBTI individual CG: 53 (52 M, 1 W) A = 72.2 IG: 72.1 CG: 72.4 C: (1) +60 years. (2) Diagnosis of insomnia. (3) Not having sleep apnea, or scores below 20 on the apnea hypopnea index. (4) Results higher than 24 on the Mini Mental Test. (5) Not having serious physical or mental pathologies.	IG: CBTI was applied in 5 sessions spread over 6 weeks including psychoeducation, sleep restriction, stimulus control, and cognitive therapy. CG: A general sleep education pro-gram structured in the same way as IG was applied.	Sleep diary and actigraphy: for 7 days per assessment. Sleep quality and quantity. PSQI: sleep quality.	Evaluation T0: pre-interv. T1: 6 months post-interv. T2: 12 months post-interv. Results Improved sleep quality in IG compared to CG. There are no significant differences between perceived and measured sleep time.	IG: 9 did not complete the interv, 8 were lost to follow-up. CG: 2 were lost to follow-up.
Perini et al. [39], 2021. Singapore	Randomized con-trolled clinical trial.	N = 127 (53 M, 74 W) IG: 65 (29 M, 36 W) CG: 62 (24 M, 38 W) A: 60.9 IG: 61.2 CG: 60.7 C: (1) Be fluent in English. (2) Have no cognitive deficiencies or score greater than or equal to 26 on the Mini Mental State Test or greater than or equal to 23 on the Montreal Cognitive Assessment. (3) Have reported sleep problems in the last month	IG: Weekly 2 h intervention for 8 weeks in which mindfulness exercises, meditation, and body scans were carried out. Afterwards, participants shared their experiences with these practices and how it had affected their sleep quality. Most of the classes contained educational content on sleep hygiene and behavioral strategies. They were taught by certified mindfulness teachers. Each participant was provided with a booklet and audios to practice mindfulness at home. CG: They received a different intervention of the same duration in which a clinical psychologist provided information on sleep biology, self-monitoring of sleep behavior, and were taught changes in habits and environment that could improve sleep quality. They also learned exercises to promote sleep and were provided with booklets and audios to practice at home.	PSQI, ISI: Sleep quality. Actigraphy and polysomnography: sleep latency and awakenings. Five-facet Mind-fulness Questionnaire, Pre-Sleep Arousal Scale, and Dysfunctional Beliefs and Atti-tudes about Sleep Questionnaire: time in bed, total time sleeping, and sleep efficiency.	Evaluation T0: pre-interv. T1: week 4 of interv. T2: post-interv. T3: 6 months post-interv. Results Both CG and IG achieved significant improvement in sleep quality. IG: achieved better results in reducing insomnia.	The sessions could not be recorded to ensure that the protocols were followed correctly and to preserve the privacy of the participants. IG: 2 did not attend the interview, 6 did not complete the sessions, 5 were lost to follow-up, 7 did not take the correct measurements. CG: 1 did not receive the interview, 3 did not complete the sessions, 26 did not take the correct measurements, 6 were lost to follow-up.
Camino et al. [40], 2022. Valencia.	Randomized con-trolled clinical trial.	N = 96 (20 M, 76 W) IG: 48 (10 M, 38 W) IGsubclinic: 23 IGmoderate: 25 CG: 48 (10 M, 38 W) CGsubclinic: 24 CGmoderate: 24 A: 72.9 C: (1) +60 years. (2) Be able to read and have no vision problems. (3) Have an ISI score be-tween 8 and 21 and at least 6 on the Pittsburgh index. (4) Not have serious physical or mental health problems. Participants were divided into 2 groups: with subclinical insomnia and with moderate insomnia. They were then randomized into CG and IG.	IG: 1 weekly session for 8 weeks lasting 1.15 h using cognitive mindfulness therapy to improve sleep quality. The first 15 min were meditation, followed by a discussion and sharing of participants’ emotions and progress during the week, then moving on to the different mindfulness sessions of autopilot, facing the challenge, conscious breathing, living in the present, letting go, thoughts are not facts, ways to take better care of myself and use what I have learned. CG: 1 weekly session for 8 weeks of films about active aging.	ISI: Participants’ perception of insomnia. PSQI: Qualitative and quantitative aspects of sleep.	Evaluation T0: pre-interv. T1: post interv. Results The intervention significantly improved sleep quality in older adults with moderate and subclinical insomnia. It also significantly improved the severity of insomnia in subjects with moderate insomnia. After the intervention, treatment was more effective in participants with moderate insomnia than in those with subclinical insomnia.	IG: 4 excluded for medical reasons, 1 did not attend all sessions. CG: 5 did not complete the second evaluation.

Note: AHI (apnea-hypopnea index), CBT-I (cognitive behavioral therapy for insomnia), C (characteristics), A (age), CG (control group), IG (intervention group), M (men), h (hour), interv (intervention), ISI (insomnia severity index), W (women), min (minutes), PSQI (Pittsburgh sleep quality index), T0 (initial or preintervention evaluation), T1/2/3 (1st/2nd/3rd post-intervention evaluation), tlf (telephone/telephone).

**Table 5 healthcare-13-01078-t005:** CASPe guidelines for clinical trials.

	Lovato et al. [32], 2014. Adelaide.	Alessi et al. [33], 2016. Los Angeles	Lovato et al. [34], 2016. Adelaide.	Bergdahl et al. [35], 2017. Uppsala.	Gallegos et al. [36], 2018. New York	McCrae et al. [37], 2018. Gainesville.	Dzierzewski et al. [38], 2019. Los Angeles.	Perini et al. [39], 2021. Singapore.	Camino et al. [40], 2022. Valencia.
Did the study address a clearly formulated research question?	Yes	Yes	Yes	Yes	Yes	Yes	Yes	Yes	Yes
2. Was the assignment of patients to intervention randomized?	Yes	Yes	Yes	Yes	Yes	Yes	Yes	Yes	Yes
3. Were all patients who entered the study accounted for at its conclusion?	Yes	Yes	Yes	Yes	Yes	Yes	Yes	Yes	Yes
4. Was blinding maintained for patients, clinicians and study personnel?	No	No	No	No	No	No	No	No	No
5. Were the study groups similar at the start of the trial?	Yes	Yes	I don’t know	Yes	Yes	Yes	Yes	Yes	Yes
6. Apart from the intervention, did each study group were treated equally?	Yes	Yes	Yes	Yes	Yes	Yes	Yes	Yes	Yes
7. Is the treatment effect reported comprehensively?	Yes	Yes	Yes	No	Yes	Yes	Yes	Yes	Yes
8. What is the accuracy of this effect?	Yes	Yes	Yes	Yes	Yes	Yes	Yes	Yes	Yes
9. Can these results be applied in your local environment or population?	Yes	Yes	Yes	Yes	Yes	Yes	Yes	Yes	Yes
10. Were all clinically important outcomes taken into account?	Yes	Yes	Yes	Yes	Yes	Yes	Yes	Yes	Yes
11. Do the benefits to be obtained justify the risks and costs?	Yes	Yes	Yes	Yes	Yes	Yes	Yes	Yes	Yes
Total	10/11	10/11	9/11	9/11	10/11	10/11	10/11	10/11	10/11

## Data Availability

Available in the references.

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
