# Peer review of "Cognitive Interventions for the Treatment of Insomnia or Poor-Quality Sleep in Community-Dwelling Older People: A Systematic Review and Meta-Analysis"

_healthcare, 2025, doi:10.3390/healthcare13091078_

Round 1

Reviewer 1 Report

Comments and Suggestions for Authors

OVERALL EVALUATION

The idea of exploring cognitive treatments for sleep in older people is really interesting, however, there are some key issues with structure and content in this paper.

SPECIFIC FEEDBACK:

Abstract

  1. The phrasing of the Results section of the abstract is wordy and hard to follow. It would be useful to re-write this to be clearer by breaking it down into smaller sentences and/or simplifying the word choice.

Introduction:

The story of the introduction feels unclear, with many brief paragraphs that highlight points that do not necessarily connect to each other or the overall picture of the paper. I would encourage the authors to re-evaluate the story of the introduction and update it accordingly. See below for some specific points.

  1. The second paragraph is about the aging population – this could be tied in better to the first paragraph or connected more so to the idea of the impact of health problems.
  2. The third paragraph specifically highlights Spain, however, this is the only time that Spain is brought up. Either I would drop this paragraph, or expand on it and tie the discussion back into this in some way.
  3. The fourth and fifth paragraph talk about age and health, however, these could be better connected to the previous paragraphs. One way might be to highlight that with the increasing aging population, there will become more strain on health care systems, and then highlight the various potential health issues.
  4. Following the previous point, the third last paragraph can then be set up better, because you can highlight that one way to improve health outcomes for older people is to explore sleep interventions. However, it would be useful to explain more explicitly why looking into nonpharmacological treatments is so important. Are pharmacological treatments less successful in older people? Are they more expensive? Are older people more resistant to that kind of treatment? Could non-pharmacological treatments also potentially improve other health outcomes?

Method:

Again, this section feels very brief at times and could benefit from some elaboration.

  1. Table 1 requires some more elaboration, specifically why people over 60 were chosen as the population (this could also be set up in the introduction if relevant).
  2. A lot of the sentences in this section are single-sentence paragraphs, many of which could be combined for easier reading.
  3. In the search strategies listed, could the authors explain why they only searched for “cognitive” and not things like “cognitive interventions”, “psychotherapy”, etc.?
  4. In the Inclusion and Exclusion Criteria section, the first sentence ends in a colon (:), but then the next paragraph is a sentence snippet “detailed in Table 3.”, please edit this. Since your inclusion and exclusion criteria are fairly short, you could replace the table with the criteria written in text. This would also allow you to elaborate on why these criteria were chosen, such as why people over 60 (coming back to my point above), and why exclude nursing homes. Especially since you actually do this kind of explanation in the paragraphs following the table when discussing randomised trials.
  5. The writing in the Quality Assessment section is unfinished and looks like it may have been a draft accidently uploaded. The first paragraph has extra line indents and looks as though some words may have been left out. The second paragraph with the bullet points could have simplified into a single paragraph with each bullet point being its own sentence. And the last paragraph looks as though the first part of the sentence is missing.
  6. The first mention of the PSQI acronym in the Analysis of Data Obtained section needs to be expanded.

Results:

The results section could have a better structure/flow to make them easier to follow.

  1. In the study selection and characteristics section, the single sentences could be combined into a single paragraph.
  2. Table 4 is difficult to read due to the spacing and the way it is set up. Either the table could be redesigned to read easier, or there also needs to be some in-text description about with this table is highlighting, such as summarising those results.
  3. It is unclear why Table 5 is in the Risk of Bias section, it seems as though it should be in the previous section. This seems reflected in the text, where the authors refer to Table 5 as talking about the CASPe guidelines, but that is presented in Table 6.
  4. The labelling of sections is unclear – 3.1 is study characteristics, 3.2 is risk of bias, but then 3.3, 3.4, 3.5, and 3.6 are all about different sleep measures. Firstly, it would be better to group the sleep measures under 3.3, and then have the unique measures separated into 3.3.1, 3.3.2, and so on. But secondly, why were they separated by sleep measure. If the focus of the paper is cognitive interventions that improve sleep, it feels like the sections should have been grouped by intervention type. Since the analysis is based on the sleep measure type, it needs to be made clearer in the results which intervention is occurring, since this might explain differences in outcomes for a particular measure.

Discussion:

  1. A foundational piece that is missing for me in this paper is more explanation on the different types of cognitive therapies in each study examined. Not all cognitive therapies are the same, and the different types are likely to impact the results differently, and this has not been discussed at all. The first paragraph of the discussion highlights some intricacies of different therapy types (which should have been set up in the introduction), but not in relation to the current results.
  2. The paragraph explaining nonpharmacological vs. pharmacological treatments should have been set up in the introduction, so this paragraph could just be calling back to that and connecting to the current results.
  3. Many of the non-pharmacological examples raised feel disjointed and not part of the overall story, this needs more set up and connection.
  4. The limitations could have much more depth – what about differences in age (do treatments work the same for a 60 year old vs. a 90 year old?), and other factors that were not explore, like gender (which only briefly was mentioned earlier in the discussion), cultural differences, whether they are still working, and so on.

Formatting

I have already highlighted some formatting issues in specific sections, but there are also inconsistencies with in-text referencing formatting (e.g., Lovato N et al.).

Reviewer 2 Report

Comments and Suggestions for Authors

Dear authors,

The study addresses a key issue in the current context of population aging, namely the quality of sleep in older adults, from a modern perspective focused on non-pharmacological interventions. The article provides well-founded arguments supporting the implementation of cognitive therapies at the level of primary care, considering both their effectiveness and cost-efficiency compared to pharmacological treatments.

However, I believe the manuscript has some shortcomings that I would like to point out, as follows:

  • Although the objective of the study is mentioned in the introduction, it is not formulated clearly, concisely, or prominently enough. I recommend it should be highlighted in such a way that readers can quickly grasp the purpose of the study;
  • The discussion section, while well documented, is very dense. I suggest restructuring it along the following lines: the effectiveness of cognitive interventions, comparison with other non-pharmacological therapies, the importance of the clinical context, and the theoretical and practical contributions of the study;
  • In the limitations section, you mentioned: “...all the studies included in this review presented a low risk of bias.” I consider it would be important to include comparative data or CASPe scores to support the claim of high methodological quality and, therefore, low risk of bias—especially since these data are presented in the results section.

I hope these suggestions will be helpful in further improving the quality and clarity of your work. I appreciate the contribution your study makes to the field and look forward to seeing it refined for publication.

Comments on the Quality of English Language
  • Some sentences are long and occasionally redundant, which affects the fluency and readability of the text.

  • Certain phrases are unclear or awkwardly constructed and would benefit from rephrasing for clarity.

  • The use of specific terms or expressions is sometimes imprecise or informal (e.g., "good sleep quality has been shown to be important" could be expressed in a more academic tone).

  • Logical connectors and transitions between ideas and paragraphs are occasionally missing, which weakens the coherence of the text.

  • Punctuation is sometimes inconsistent (e.g., incorrect spacing between words and punctuation marks, such as “interv .” or “be er”).

Reviewer 3 Report

Comments and Suggestions for Authors

The prevalence of sleep disorders is a very serious and significant problem. In this regard, the study of non-drug methods for correcting sleep disorders is a relevant topic.

The Materials and Methods section is very large: the authors are advised to shorten it, as this is not the main section of the manuscript.

In the Introduction section, special attention should be paid to the description of existing non-drug cognitive methods for correcting sleep disorders, and to define and characterize these methods.

It is also necessary to indicate the pros and cons of each method in comparison with drug methods for correcting sleep disorders.

In the Results section, in addition to methods for correcting sleep disorders and insomnia, methods for studying sleep disorders are given. Due to the fact that the title of the manuscript only indicates cognitive methods for correcting sleep disorders, this aspect should be fully disclosed in the manuscript.

In the Conclusion section, the conclusions should be formulated more clearly, indicating the possibility of further application of the obtained results in clinical practice.

Reviewer 4 Report

Comments and Suggestions for Authors

“Cognitive Interventions for the Treatment of Insomnia or Poor Quality of Sleep in Older People in the Community: A Systematic Review and Meta-Analysis”( healthcare-3549177)

This systematic review aimed to determine the effectiveness of different cognitive interventions for the treatment of insomnia or poor sleep quality in community-dwelling older people. Nine clinical trials were retained for meta-analysis. The results reveal that cognitive interventions have been found to be safe and useful for the treatment of insomnia and poor sleep quality in older people. This topic is interesting and important. However, some concerns appeared after reading the whole manuscript.

  1. This topic has already been solved by some papers, which are missing in the current manuscript. This fact combined with some missing important reviews published recently made me wonder if the authors really get the latest up-to-date of this topic.

McLaren, D. M., Evans, J., Baylan, S., Smith, S., & Gardani, M. (2023). The effectiveness of the behavioural components of cognitive behavioural therapy for insomnia in older adults: A systematic review. Journal of sleep research, 32(4), e13843.

Huang, K., Li, S., He, R., Zhong, T., Yang, H., Chen, L., ... & Jia, Y. (2022). Efficacy of cognitive behavioral therapy for insomnia (CBT-I) in older adults with insomnia: a systematic review and meta-analysis. Australasian Psychiatry, 30(5), 592-597.

  1. The more serious concerns lie in the methodology. The search strategy would be problematic since the two systematic reviews mentioned above focused on one specific kind of cognitive interventions (cognitive behavioral therapy). McLaren identified 15 studies and Huang identified 14 studies, which are much more than only 9 trails identified in the current investigation. It is surprising to find that the “insomnia” in the title is missing in the search strategy, which would make the literature search incomplete.

  1. The prevalence of insomnia and sleep disturbance should be updated.

  1. For the exclusion criteria:“Studies with more than 10 years of experience.” Which date did you account for? November 2023?or July 2024?

  1. All the funnel plots seem to be odd.
Comments on the Quality of English Language

 The English could be improved to more clearly express the research.

Round 2

Reviewer 1 Report

Comments and Suggestions for Authors

OVERALL EVALUATION

Thank you for addressing the feedback points provided, many of these points have been adequately addressed. However, there are several points that need further editing. See below for those specific points.

SPECIFIC FEEDBACK:

Introduction:

  1. Paragraphs 2, 3, and 4 can all be combined into a single paragraph now.
  2. Paragraphs 5 (active aging) and 6 (sleeps problems in older people) still need adjustment to connect to the surrounding text.
    • The story of the introduction would flow better if you put paragraph 6 first and edit the first sentence from “As previously mentioned, sleep problems…” to something like “In particular, sleep problems…” which would flow better from paragraph 4.
    • At the end of paragraph 6, I would then move the first sentence from paragraph 7 “This issue arises because, over the years, sleep patterns and circadian rhythms slow down due to the presence of steroid hormones, body temperature, and melatonin, which significantly impacts women (8,16)” into this paragraph.
    • Then the next paragraph would be your paragraph on active aging, but I would rephrase this paragraph to fit in better with the surrounding paragraphs by editing the first sentence from “To prevent and/or delay these changes, the term “active aging” is increasingly being used” to something like “As highlighted by the World Health Organisation, healthy aging is an important way to improve the quality of life of older people…” – focus the paragraph on being about why it is important to encourage healthy aging (could even talk about the economic costs of aging, etc.), and the ways healthy aging is encouraged.
    • This would then lead clearly into the solutions for improving sleep, which would be where you can combine the rest of paragraph 7 (main solution for insomnia) with paragraph 8 (various therapies).
    • Also, make it clear in the sentence “Furthermore, it is well known that the use of medication generates health expenditures, as well as dependence (especially benzodiazepines), and has excessive adverse effects (18)” whether this is specific to older people or not.
  3. The specificity of “people over 60 years of age who live in the community” is phrasing at the end of the introduction still doesn’t quite work. Either you need further context here specifically, or just leave the phrasing as “older people” and then just detail the specifics in the method.

Method:

  1. Following the point above, whilst I appreciate that the authors have provided some explanation for the choice of over 60s, this paragraph needs further editing. The authors reference the WHO and the United Nations, but the references provided only refer to the United Nations, and then link for the United Nations reference does not work. It would be better to be transparent and say something like “The age cut-off point was set at people over the age of 60 years old, following the cutoff used by the WHO and the United Nations when discussing older people…” and provide appropriate references.
  2. The added context for the exclusion of nursing homes needs more elaboration. Add references to support why there a need to evaluate the effectiveness of interventions in older people with more independence, and be clearer about “making it difficult to extrapolate findings to the general population” – do you mean that it makes it difficult to extrapolate findings to older people more broadly.
  3. Now that you have added “who do not suffer from other serious pathologies” to the inclusion criteria, this also warrants an explanation in the text.

Results:

  1. I think you confused my point about the clarity of Table 4 with Table 5. Table 4 in the original manuscript was “Comparison and synthesis of the intervention group and comparator group”, which is now removed from the current manuscript. Table 5 in the original manuscript is Table 4 in the current manuscript, and provides the “main characteristics of the studies included in this systematic review”. I appreciate the editing of the new Table 4 to be clearer, but can the authors explain the removal of the old Table 4 and whether this information has been provided elsewhere?

Discussion:

  1. The first paragraph reiterates cognitive interventions in sleep, but some of the information could be moved to the introduction and better set up there. For example, the final sentence about mindfulness would be better in the introduction and then just briefly called back to here.
  2. The opening sentence of paragraph 2 is too long and doesn’t make sense. I appreciate the addition to the start of this paragraph, but it would be better if this was pulled apart and elaborated further. The first sentence could state that this review highlights that non-pharmacological treatments play a beneficial role in improving sleep quality in older people. Then the second sentence can state that this is important because, compared to pharmacological treatments, non-pharmacological treatments are not utilised as often due to lack of knowledge and awareness about their benefits, including not interacting with medications, etc.
  3. In my point about the non-pharmacological examples being out of place, providing in-text linking words/phrases doesn’t improve this issue. Paragraphs 4, 5, and 6 list different statements that don’t necessarily fit with the story. It is unclear what the main purpose of these paragraphs are. The first sentence of paragraph 4 talks about combining non-pharmacological therapies, but then some of the examples raised are not combined examples. If the purpose is to outline different therapy types that have been effective, then this needs to be the set-up, and an explanation as to why this is relevant is needed (is it further evidence that there are other solutions as opposed to pharmacological treatments? These weren’t explored in the review, should future research consider this?).
  4. The following paragraph on educating health professionals also needs more connection. Why are nursing staff specifically pointed out? How does education about sleep health for health professionals fit into the story of this review? Is it a way to educate health professionals about treatments other than pharmacology for sleep problems?
  5. As suggested in my original review, the limitations could have much more depth. The brief additions do not add enough to this section.
    • For example, age needs to be teased apart and discussed further. It would be useful to mention the limitation of making the cut-off 60 (are there differing definitions of aging around the world, particularly where life expectancies are different?), and the possible differences within the age bracket of over 60 (as I mentioned previously, 60 vs 90 are very different life stages).
    • The exclusion of nursing homes could be a big limitation – what are the numbers of older people in nursing homes? If the number is high, then you might be leaving out a huge portion of older people. Would they still be able to benefit from cognitive therapies?
    • More elaboration about the differences between therapy types, i.e., the fact that this review did not tease apart the effect of different lengths of therapy, and type of therapy (mindfulness, CBT-I, combinations, etc., particularly if there are any references you can add to suggest that a particular type might be more suited).
    • More elaboration of gender – I would move the sentence earlier in the discussion that mentions gender to here and elaborate about the differences in poor sleep and whether there are similar improvements for men and women or not based on the literature.
  6. In the conclusion, the authors highlight the cost-effectiveness of cognitive interventions – this point needs to be set up in the introduction when comparing to pharmacology, and also needs to have some elaboration on how it is more cost-effective. If cognitive interventions like CBT-I need to be administered by psychologists, this would incur the cost of seeing a psychologist for many sessions, which may actually be more expensive than getting a prescription from your doctor.

Formatting

The reference numbers are out of sync – for example, there are two 18s.

Reviewer 2 Report

Comments and Suggestions for Authors

Dear Authors,
Thank you for taking my suggestions into consideration. I believe your research paper is now greatly improved.
Wishing you success in your future research!

Author Response

Dear revisor, thank you very much for your suggestions and help in improving the quality of our manuscript.

Best regards, the authors.

Reviewer 3 Report

Comments and Suggestions for Authors

The authors have significantly revised the manuscript, as a result of which the manuscript began to look stronger and more significant.

Author Response

(The authors gave the same response as above.)

Reviewer 4 Report

Comments and Suggestions for Authors

Thanks for the revisions. It is regret to find that the revised version was not significantly improved. And some critical concerns in the previous round remain.

Round 3

Reviewer 4 Report

Comments and Suggestions for Authors

Thanks for the revisions again. However, this revised version still did not improved much. The authors did not address the concerns well which I raised in the first round.
